# Using the Oral Assessment Guide to Predict the Onset of Pneumonia in Residents of Long-Term Care and Welfare Facilities: A One-Year Prospective Cohort Study

**DOI:** 10.3390/ijerph192113731

**Published:** 2022-10-22

**Authors:** Masahiro Yamanaka, Kanetaka Yamaguchi, Masumi Muramatsu, Hiroko Miura, Morio Ochi

**Affiliations:** 1Division of Fixed Prosthodontics and Oral Implantology, Department of Oral Rehabilitation, School of Dentistry, Health Sciences University of Hokkaido, Ishikari-gun 061-0293, Japan; 2School of Nursing, Sapporo City University, Sapporo 060-0011, Japan; 3Division of Disease Control and Epidemiology, School of Dentistry, Health Sciences University of Hokkaido, Ishikari-gun 061-0293, Japan

**Keywords:** dental, OAG, pneumonia, residents of long-term care and welfare facilities

## Abstract

Appropriate oral health care, depending on oral assessments, reduces the onset of pneumonia. However, the relationship between risk evaluation using an oral assessment tool and pneumonia in residents of long-term care facilities has not been fully elucidated. In the present study, we aim to examine the relationship between the total scores of the Oral Assessment Guide (OAG) and the incidence of pneumonia after a one-year baseline study of residents in long-term care facilities. The settings for sampling include nine long-term care facilities in Hokkaido. At baseline, there were 267 study subjects. A total of 72 individuals dropped out and 11 individuals met the exclusion criteria. Therefore, the subject sample included 184 individuals. Among the subjects included in our analyses, eight individuals developed pneumonia (six males and two females). A multiple logistic regression analysis was performed with the risk factors for developing pneumonia. Evaluations were performed based on the odds ratio (OR) and 95% confidence interval (CI). We observed that the OR for pneumonia onset was 2.29 (CI: 1.27–4.14) after being adjusted for pneumonia risk factors. Therefore, it was suggested that the total scores of the OAG could be used to screen for the risk of pneumonia onset in residents of long-term care and welfare facilities.

## 1. Introduction

The average life expectancy in Japan is the highest in the world at 84.3 years of age [1]. It has been reported that elderly individuals aged 65 years or older and individuals with concurrent non-communicable diseases (NCDs) are at a high risk of death as a result of pneumonia [2]. According to the Japanese vital statistics, in 2021, pneumonia (6.4%) was the fourth leading cause of death in the Japanese population, followed by senescence (5.1%) as the fifth leading cause, and aspiration pneumonia (3.5%) as the sixth leading cause. This indicates that pneumonia remains a major problem in elderly individuals [3]. Countermeasures against pneumonia should be promoted in Japan, where society is aging at an alarming rate.

Furthermore, residents of long-term care facilities are at a high risk of pneumonia onset among elderly individuals with pneumonia [4,5,6,7,8]. It has been reported that the incidence of and hospitalization for pneumonia in long-term-care-facility residents are approximately 6–10- and 30-fold higher, respectively, than those of elderly individuals living in the community [9]. The mortality rate of elderly patients with pneumonia is reported to be 12.2% in the community-dwelling elderly, 19.6% in nursing-care hospital inpatients, and 40.3% in residents of long-term care facilities [10]. Therefore, countermeasures against pneumonia are urgently needed for elderly residents in long-term care facilities.

While residents in long-term care facilities are at a high risk of developing pneumonia [4,5,6,7,8], providing appropriate oral health care to such individuals reduces the onset of pneumonia [11,12,13,14]. Poor oral health can lead to respiratory complications. It has been reported that patients with periodontal disease have a four-fold increased risk of requiring hospitalization, a six-fold increased risk of requiring ventilator use, and a seven-fold increased risk of death from complications that worsen the condition of COVID-19 [15]. COVID-19 showed significant associations with plaque scores (odds ratio (OR), 7.01; 95% confidence interval (CI), 1.83 to 26.94), gingivitis (OR, 17.65; 95% CI, 5.95 to 52.37), and severe periodontitis (OR, 11.75; 95% CI, 3.89 to 35.49) [16].

Thus, oral health care can help to prevent pneumonia occurrence in residents of long-term care facilities. In particular, improving oral assessments promotes oral health care for patients [17], and it is therefore important to perform oral evaluations using an oral assessment tool. In Japan, oral assessment tools include the Oral Health Assessment Tool (OHAT) [18], the Oral Assessment Guide (OAG), and the Revised Oral Assessment Guide (ROAG) [19,20,21]. Each institution uses a different oral assessment tool. The OHAT is commonly used in dentistry, whereas the OAG and ROAG are commonly used in hospitals. We believe that if residents in long-term care facilities could receive oral health care based on a consistent oral health evaluation, regardless of their location changing due to admissions and discharges, it would be effective in preventing the onset of pneumonia. However, the only report on the relationship between risk evaluation using an oral assessment tool capable of a consistent oral health evaluation and pneumonia is of inpatients aged 65 years or older using the ROAG [22]. The relationship between risk evaluation using an oral assessment tool and pneumonia for residents in long-term care facilities has not yet been fully elucidated.

Therefore, in the present study, among the OHATs, we focus on the OAG, which has proven to be reliable and valid [23,24,25,26]. Moreover, this tool is used in basic nursing education in Japan, is the recommended tool for oral evaluations by nurses [27], and is useful for sharing information about oral health among different medical teams [24].

The OAG was developed by Eilers et al. for patients receiving cancer treatment [28]. The OAG consists of eight assessment categories: voice, swallowing, lips, tongue, saliva, mucous membrane, gingiva, and teeth/dentures. Each category is expressed in three stages. The OAG score is the sum of the scores for each category; each category is scored from 1 to 3. The best possible score of 8 points is indicative of good oral health; the worst possible score of 24 points is indicative of poor oral health. It has been reported that the OAG is suitable for evaluating daily life activities [29] and the oral health of elderly individuals [30].

Therefore, as an indicator of oral health assessment that is shared by the departments of medical care and dentistry, we propose the following research question: Can the OAG serve as an OHAT capable of evaluating the risk of pneumonia onset? In order to answer this research question, we examine the relationship between the total scores of the OAG and a one-year incidence of pneumonia in residents of long-term care facilities in Japan, before the onset of the COVID-19 pandemic.

## 2. Materials and Methods

The study design was a one-year prospective cohort study.

### 2.1. Subjects and Recruitment Method

This study included nine long-term care facilities and used the method of snowball sampling, where the purpose of the study was explained in advance, and consent and cooperation for participation in the study were obtained from the nine facilities, located in Hokkaido, Japan. In addition, direct surveys by facility visitation were conducted in Japan before the COVID-19 pandemic (from July 2018 to February 2020).

In order to minimize the influence of cognitive function, we sampled subjects using the degree of life independence, which is an index that evaluates the degree of daily living due to dementia [31,32].

The subject selection criteria included elderly individuals with dementia. The degree of life independence was evaluated as ≥grade III for those who had a diagnosis of dementia, according to the guidelines of the Ministry of Health, Labour, and Welfare [33]. The exclusion criteria included individuals who had undergone dental therapeutic intervention during the year after the baseline survey in order to exclude the improvement of the OAG score through dental treatment.

### 2.2. Investigators and Ethical Considerations

Two dentists from the Health Sciences University of Hokkaido (with 6 and 3 years of clinical experience, respectively) conducted surveys at the target long-term care facilities. Furthermore, prior to the surveys, individuals in charge at the facilities were informed of this study and consent for study cooperation was obtained. The researchers and individuals in charge at the participating facilities explained the study to the target individuals, and subjects who consented to participate in the study signed the study consent form. Individuals who were unable to sign on their own were asked to have a legal representative sign on their behalf. On the day of the survey, the subjects and legal representatives were given a consent withdrawal form and informed that they could withdraw their consent at any time, and their intention to participate in the study was reconfirmed. Furthermore, the study was conducted with the approval of the ethical review board of the Health Sciences University of Hokkaido (approval number: 178).

### 2.3. Test Items

According to a previous study [9,34,35], subjects, age, gender, and number of oral medications were selected as risk factors for pneumonia onset. Additionally, cerebrovascular disease, cardiovascular disease, diabetes mellitus [36], and oral health care [11,12,13,14] were obtained from medical records kept in files at each facility. Furthermore, two dentists used the OAG to evaluate and score subjects’ oral cavities. An evaluation using the OAG was performed based on an existing photo of the oral cavity. The consistency among the examiners was evaluated using the kappa coefficient. The calibration among examiners was performed until the kappa coefficient reached at least 0.9. The obtained data were anonymized to conceal each subject’s personal information, such as the subject’s name, and the special care they required. Moreover, precautions were taken to prevent the disclosure of personal information, where each facility managed a reference table that prevented the researchers from identifying the study subjects.

### 2.4. Sample Size Calculation

We planned a study of matched sets of subjects who did or did not develop pneumonia. We calculated the sample size using power and sample size calculations. We could not obtain the condition values for the calculation of the sample size from previous studies, so we estimated and substituted them into the calculation formula. Developing pneumonia among the controls was 0.1, and the correlation coefficient for exposure between matched experimental and control subjects was 0.4. If the true odds ratio for failure in experimental subjects relative to control subjects was 2, we needed to study 266 experimental subjects with 1 matched control per experimental subject to be able to reject the null hypothesis that the odds ratio equalled 1 with a probability (power) of 0.9. The type-I-error probability associated with the test of this null hypothesis was 0.05.

### 2.5. Statistical Analysis

The normality of the scale data was tested and confirmed using the Kolmogorov–Smirnov test. Then, a logistic regression analysis was performed, with pneumonia onset after 1 year of the baseline survey as the dependent variable; the total OAG score as the explanatory variable; and age, gender, number of oral medications, presence or absence of a medical history (cerebrovascular disease, cardiovascular disease, and diabetes), and the presence or absence of oral health care as the modulator variables. Prior to the logistic regression analysis, the correlation between the moderator variables was verified using the Spearman’s correlation coefficient, and the presence or absence of multicollinearity was examined. In model 1, we evaluated a logistic single regression analysis of dependent and explanatory variables; in model 2, age and gender were inserted into model 1. In model 3, a multiple logistic regression analysis was performed with the number of oral medications, presence or absence of a medical history, and the presence or absence of oral health care inserted into model 2. Evaluations were performed based on the odds ratio (OR) and 95% confidence interval (CI). Furthermore, the conformity of the regression model was examined using the Hosmer–Lemeshow and chi-squared tests.

We used IBM SPSS Statistics^®^ (version 24.0, IBM Corporation, Somers, NY, USA) for the statistical analysis.

## 3. Results

At baseline, there were 267 study subjects (Table 1). From the time of the baseline examination at each facility until 1 year later, 72 individuals dropped out (Figure 1, Appendix A: Appendix A). Furthermore, 11 individuals met the exclusion criteria (Figure 1, Appendix A: Appendix A), and thus were excluded from the analyses. Therefore, the subject sample included 184 individuals (31 men, mean age = 85.0 ± 8.3 years; 153 women, mean age = 87.9 ± 6.1 years; Figure 1). In the sample from nine facilities at baseline, there was no significant difference in the male-to-female ratio; however, an interfacility difference was observed in the dropout rate. After 1 year, there was no significant difference between the male-to-female ratio in the sampling from the nine facilities (Table 2). Among the subjects included in our analyses, eight individuals developed pneumonia (six males and two females), with differences observed in the number of cases between facilities (Appendix A: Appendix A).

Non-normality was observed in all scale data items (Appendix A: Appendix A); no correlation coefficient between the explanatory variables exceeded 0.8; and we confirmed that there was almost no multicollinearity (Table 3).

The statistical analysis results for the total OAG scores reveal the following:

OR: 1.56, 95% CI: 1.16–2.11, and *p* < 0.01 in model 1; OR: 1.57, 95% CI: 1.11–2.22, and *p* < 0.05 in model 2; and OR: 2.29, 95% CI: 1.27–4.14, and *p* < 0.01 in model 3 (Table 4).

Furthermore, after examining the conformity of the regression model using the Hosmer–Lemeshow test, we observed that the *p*-value was 0.05 or greater in all the analysis models, indicating that conformity was maintained in all models (Appendix A: Appendix A). After examining the significance of the regression model using the chi-squared test, we observed that the *p*-value was less than 0.05 in all the analysis models, confirming the significance of the regression equation (Appendix A: Appendix A).

## 4. Discussion

The OR of pneumonia onset was 2.29 after being adjusted for pneumonia risk factors. The results suggest that the OAG could be used as an explanatory variable for pneumonia onset in residents of long-term care and welfare facilities in Hokkaido, Japan.

It has been reported that residents of long-term care facilities are at a higher risk of pneumonia onset than the elderly who live in the community [4,5,6,7,8,9]. Accordingly, the subjects of the present study were at a high risk of pneumonia onset; however, there was only a slight onset of pneumonia in this population.

Previous studies on pneumonia in the elderly population reported that the risk factors included comorbidity (heart disorder, cerebrovascular disease, lung disease, diabetes mellitus, and malignant tumours) [34], gender (male) [9,35,37], smoking habits [9,35,37], nasogastric tube feeding [36,38,39], swallowing difficulty [36,38,39], use of sedatives [36,38,39], and analgesics [36,38,39]. In addition, it was reported that oral health care reduced the incidence of pneumonia [11,12,13,14]. We also observed that gender (male) and cerebrovascular diseases were risk factors for pneumonia onset in the subjects in the present study.

The following three points were the strengths of the present study:

Firstly, we demonstrated the relationship between the total scores of the OAG and the onset of pneumonia in residents of long-term care and welfare facilities by a prospective cohort study. Prior studies of OAG included inpatients [40,41] and healthy community-dwelling elderly individuals [30], whereas the present study was unique in that it included residents of long-term care and welfare facilities. Furthermore, in terms of the relationship between oral assessment tools and pneumonia, the ROAG reported pneumonia onset in inpatients aged 65 years or older with a history of pneumonia [22]. The mean subject age in the present study was 87.7 years, which was higher than in the previous studies, suggesting a relationship between pneumonia onset occurring after 1 year of the subject being a resident in long-term care and welfare facilities, which is a new observation that has not been mentioned in the previous reports.

Secondly, we suggested that OAG can be an explanatory variable for pneumonia onset independently of pneumonia risk factors reported in the previous studies. The results of the present study are expected to further promote the implementation of oral health assessment using the OAG in the field of nursing care. The OAG is used in studies examining the effectiveness of oral care protocols and in evaluations of oral health care [24,25,42,43,44,45,46]. Based on the results of the present study, screening for the risk of pneumonia onset using the OAG tool could be performed based on oral health evaluations. The results of the present study could be reflected in the evaluation of oral health care protocols. Therefore, the OAG may be able to unify care from screening for the onset of pneumonia up to oral healthcare assessments.

Furthermore, residents of long-term care facilities are at a high risk of pneumonia onset. Assessments using the OAG are simple to perform and do not require any specific equipment, making them useful in nursing and long-term care. Based on the results of the present study, if oral health could be assessed using the OAG in long-term care settings, it could be collaboratively used by departments of medical care, dentistry, and nursing care to reduce the risk of pneumonia.

Thirdly, we confirmed the conformity of a multiple logistic regression analysis and the significance of the regression equation. Therefore, the results of the present study are statistically supported, and we can conclude that the reliability and validity of the results are ensured.

The following two points were limitations of the present study:

Firstly, the required sample size was not met. The number of cases of pneumonia onset was small, and a sampling bias among the facilities was observed due to non-randomize sampling. Our study was not able to control for dropout; therefore, dropout could cause a selection bias. In future studies, we should aim to increase the target sample size by recruiting new subjects. Countermeasures were implemented to prevent the spread of COVID-19 infection; deaths from pneumonia unrelated to COVID-19 decreased to 78,445 in 2020 and 73,190 in 2021 [3,47]. COVID-19 countermeasures considerably changed the environment surrounding medical care, causing delays in the provision of medical services and a decrease in the amount of available information on other diseases [1]. It is difficult to statistically adjust for the impact of COVID-19 because it cannot be determined whether COVID-19 onset was primary or secondary to pneumonia onset. Therefore, we could not include it for analysis in the current data set.

Secondly, not all risk factors for pneumonia reported in the previous studies were adjusted. All the subjects in the present study were able to eat, and no sedatives or analgesics were administered. Furthermore, drugs used to treat dementia symptoms, urinary incontinence, depression, pain, and insomnia increased the risk of pneumonia [5,48]. However, the subjects of the present study were elderly individuals with dementia who had an independence in their daily life of grade III or lower according to the Ministry of Health, Labour, and Welfare guidelines. Few subjects were taking antipsychotics or anticholinergics. It was reported that the OAG tool is affected by stomatitis pain in patients with breast cancer [40] as well as by the nutritional status and weight loss of patients with HIV [41]. It has been suggested that pain and nutritional status alter the state of the oral cavity; however, no subjects complained of pain, and all the subjects were able to eat meals with no weight loss observed.

Vaccination against Streptococcus pneumoniae before admission is recommended as a pneumonia prevention strategy for residents of long-term care facilities [49]. However, there are no data on this subject, and it has not been examined. Accordingly, we were unable to conduct a statistical analysis on these moderator variables.

## 5. Conclusions

Despite the limitations of this study, the OR of pneumonia onset was 2.29 after adjusting for pneumonia risk factors. This suggested that the OAG could be used as an explanatory variable and a screen for pneumonia onset in residents of long-term care and welfare facilities in Hokkaido, Japan before spreading COVID-19.

## Figures and Tables

**Figure 1 ijerph-19-13731-f001:**
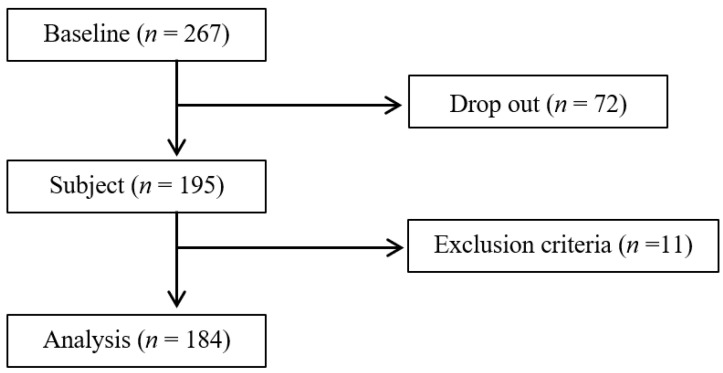
Subject diagram.

**Table 1 ijerph-19-13731-t001:** Subject information at baseline.

Long-Term Care and Welfare Facilities	Gender		Age	Medicine	OAG Total Score
	Female	Male	Mean (Min–Max)	Mean (Min–Max)	Mean (Min–Max)
1	23	6	84.93 (73–99)	8.97 (0–16)	10 (8–15)
2	34	4	87.08 (70–97)	7.62 (2–20)	10 (8–18)
3	33	7	85.53 (61–99)	7.78 (1–16)	11 (8–16)
4	32	8	89.65 (77–104)	4.48 (0–14)	10 (8–14)
5	16	4	89.25 (80–99)	8.15 (2–14)	11 (8–15)
6	30	9	88.03 (74–101)	5.21 (0–13)	11 (8–17)
7	17	2	88.95 (83–97)	7.79 (2–15)	11 (9–13)
8	26	4	89.80 (73–100)	6.17 (0–12)	10 (8–15)
9	7	5	86.42 (69–99)	10.33 (1–17)	13 (8–18)

**Table 2 ijerph-19-13731-t002:** Subject information at one year after baseline.

Long-Term Care and Welfare Facilities	Gender		Age	Medicine	OAG Total Score
	Female	Male	Mean (Min–Max)	Mean (Min–Max)	Mean (Min–Max)
1	12	3	85.80 (75–96)	7.77 (1–15)	11 (8–15)
2	26	2	87.07 (71–98)	7.11 (0–15)	9.5 (8–16)
3	25	6	85.90 (62–100)	6.58 (1–14)	10 (8–17)
4	18	6	90.75 (81–105)	7.60 (2–13)	9.5 (8–14)
5	14	2	89.93 (81–96)	6.88 (2–14)	10 (8–13)
6	27	8	89.00 (76–102)	3.53 (0–9)	10 (8–17)
7	12	2	89.78 (84–98)	6.71 (3–12)	9 (8–18)
8	19	4	90.52 (74–99)	5.55 (0–11)	12 (8–19)
9	6	3	87.44 (70–100)	10.11 (2–16)	10 (8–19)

**Table 3 ijerph-19-13731-t003:** Correlation matrix.

	OAG	Age	Gender	Medicine	Present Teeth	Oral Care by Dental Hygienist	Cardiovascular Disease	Cerebrovascular Disease	Diabetes Mellitus
OAG	1	−0.179	0.401	−0.119	0.049	−0.262	0.195	0.532	0.164
Age	−0.179	1	0.041	−0.022	−0.035	−0.095	−0.036	−0.11	−0.26
Gender	0.401	0.041	1	0.08	0.019	−0.094	−0.024	0.317	0.397
Medicine	−0.119	−0.022	0.08	1	−0.171	0.106	−0.164	−0.212	0.099
Present teeth	0.049	−0.035	0.019	−0.171	1	−0.278	0.266	−0.087	−0.002
Oral care by dental hygienist	−0.262	−0.095	−0.094	0.106	−0.278	1	−0.223	0.016	0.044
Cardiovascular disease	0.195	−0.036	−0.024	−0.164	0.266	−0.223	1	0.07	−0.255
Cerebrovascular disease	0.532	−0.11	0.317	−0.212	−0.087	0.016	0.07	1	0.016
Diabetes mellitus	0.164	−0.26	0.397	0.099	−0.002	0.044	−0.255	0.016	1

**Table 4 ijerph-19-13731-t004:** Multiple logistic regression analysis.

		Model 1	Model 2	Model 3
		OR (95%CI)	OR (95%CI)	OR (95%CI)
** OAG		1.56 (1.16–2.11)	1.57 (1.11–2.22)	2.29 (1.27–4.14)
Age			0.96 (0.86–1.06)	0.95 (0.84–1.06)
* Gender	Female		1	1
	Male		9.02 (1.51–53.95)	23.06 (2.06–258.28)
Medicine				1.03 (0.79–1.35)
Present teeth				1.00 (0.91–1.10)
Oral care by dental hygienist	No			1
	Yes			1.01 (0.14–7.57)
Cardiovascular disease	No			1
	Yes			0.90 (0.12–6.96)
* Cerebrovascular disease	No			1
	Yes			15.45 (1.37–174.38)
Diabetes mellitus	No			1
	Yes			3.25 (0.18–58.64)

* *p* < 0.05, ** *p* < 0.01. Model 1: Simple regression analysis. Model 2: Model 1 + age and gender. Model 3: Model 2 + medicine, present teeth, oral care by dental hygienist, cardiovascular disease, cerebrovascular disease, and diabetes mellitus.

## Data Availability

The data cannot be publicly shared because no informed consent was provided by the participants for open data sharing.

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
