# Peer review of "Using the Oral Assessment Guide to Predict the Onset of Pneumonia in Residents of Long-Term Care and Welfare Facilities: A One-Year Prospective Cohort Study"

_ijerph, 2022, doi:10.3390/ijerph192113731_

Round 1
Reviewer 1 Report
there is such a significant amount of non relevant information to begin with in the introduction, and a lot of injust English words, that I did not read beyond the introduction. As it is, for me this article cannot be published before a big review of relevant information and grammar/English check
Author Response
(Referee 1): There is such a significant amount of non-relevant information to begin with in the introduction, and a lot of injust English words, that I did not read beyond the introduction. As it is, for me this article cannot be published before a big review of relevant information and grammar/English check
We thank you for your thoughtful suggestions.
In the introduction, we have revised the wording to be clear and straightforward. We have also deleted expressions that deviated from the main purpose.
We had done some English editing before submission, but it did not seem to be up to a high enough standard.
So we had the English proofread again by Language Editing Services (https://www.mdpi.com/authors/english).
I look forward to working with you to move this manuscript closer to publication in the IJERPH.
Reviewer 2 Report
The authors aimed to study the impact of Oral Assessment Guide score on the risk of developing pneumonia in long-term care facility residents through a multiple logistic regression analysis. The results confirm a significant impact of oral health conditions on this respiratory complication, despite the whole study suffering from a great risk of being underpowered. The authors should properly acknowledge and discuss this point prior to receiving further consideration.
Introduction
- I would recommend synthesising introduction to render it more fluid and go straightforwardly to the point.
- Meta-analyses of epidemiological studies indicated that periodontitis subjects are more likely to experience a more severe course of COVID-19 (doi: 10.1177/00220345221104725). It would be important to provide these epidemiological numbers to highlight the impact of worsened oral health on respiratory complications.
- Please, change the aim statement as follows: Can the OAG serve as an OHAT capable of evaluating the risk of pneumonia onset? In order to answer this research question, we examined the relationship between the total scores of the OAG and the 1-year incidence of pneumonia in residents of long-term care facilities in Japan, before the onset of COVID-19 pandemics.
M&M
- Please, specify how the long-term care facilities were selected among the others. It is also unclear how the subjects were selected within each facility. Was it the result of a random sampling or simply included subjects were the ones meeting inclusion criteria and consenting to participate?
- What about selecting only 'elderly individuals with dementia'? Cognitive impairment has been robustly associated with bad oral health and this may represent a possible confounder in the analysis (doi: 10.3390/ijerph18136823)
- Line 120-122, please clarify this sentence.
- Since many readers are not familiar to OAG, please clarify which are the variables included in this score in materials and methods.
- Line 145-146, I don't understand whether OAG values were assessed using intraoral photographs. Please, clarify this point.
- Please, report also how descriptive statistics were carried out.
- Was a sample size calculation performed prior to subjects enrolment? I think that this point has a major impact on the validity and robustness of the conclusions.
Results
- What reasons the drop-out were imputable? How did the authors manage a 25% drop-out rate in the analysis?
- Please, provide descriptive statistics on the distribution of OAG scores among included subjects.
- In general, the authors should make a greater effort to present results and tables in a friendlier format. This is too messy. In addition, Table 3 and 4 are more suitable for supplementary material than the main text.
Author Response
(Referee 2): The authors aimed to study the impact of Oral Assessment Guide score on the risk of developing pneumonia in long-term care facility residents through a multiple logistic regression analysis. The results confirm a significant impact of oral health conditions on this respiratory complication, despite the whole study suffering from a great risk of being underpowered. The authors should properly acknowledge and discuss this point prior to receiving further consideration.
Introduction
- I would recommend synthesizing introduction to render it more fluid and go straightforwardly to the point.
Thank you for your advice. I deleted lines 33 through 44. I also changed the sentences.
- Meta-analyses of epidemiological studies indicated that periodontitis subjects are more likely to experience a more severe course of COVID-19 (doi: 10.1177/00220345221104725). It would be important to provide these epidemiological numbers to highlight the impact of worsened oral health on respiratory complications.
Thank you for teaching us this reference. I added the following sentence and reference in lines 53 and 352.
Line53: Poor oral health can lead to respiratory complications. It has been reported that patients with periodontal disease have a 4-fold increased risk of requiring hospitalization, a 6-fold increased risk of requiring ventilator use, and a 7-fold increased risk of death from complications that worsen the condition of COVID-19 [15]. COVID-19 showed significant associations with plaque scores (odds ratio (OR), 7.01; 95% confidence interval (CI), 1.83 to 26.94), gingivitis (OR, 17.65; 95% CI, 5.95 to 52.37), and severe periodontitis (OR, 11.75; 95% CI, 3.89 to 35.49) [16].
Line352:15.Baima G, Marruganti C, Sanz M, Romandini M. Periodontitis and COVID-19: Biological Mechanisms and Meta-analyses of Epidemiological Evidence.J Dent Res.2022(doi: 10.1177/00220345221104725)
Line354:16. Pradeep S Anand, Pranavi Jadhav, Kavitha P Kamath, Salavadi Revanth Kumar, Sandapola Vijayalaxmi, Sukumaran Anil. A case-control study on the association between periodontitis and coronavirus disease (COVID-19): J Periodontol.2022 (doi: 10.1002/JPER.21-0272.)
- Please, change the aim statement as follows: Can the OAG serve as an OHAT capable of evaluating the risk of pneumonia onset? In order to answer this research question, we examined the relationship between the total scores of the OAG and the 1-year incidence of pneumonia in residents of long-term care facilities in Japan, before the onset of COVID-19 pandemics.
Thank you for your advice. I changed the wording as you pointed out.
Line 88-92: Can the OAG serve as an OHAT capable of evaluating the risk of pneumonia onset? In order to answer this research question, we examined the relationship between the total scores of the OAG and a 1-year incidence of pneumonia in residents of long-term care facilities in Japan, before the onset of the COVID-19 pandemics.
Material & Methods
- Please, specify how the long-term care facilities were selected among the others. It is also unclear how the subjects were selected within each facility.
We added the following text to the subjects and recruitment method section.
” This study included nine long-term care facilities and used the method of snow-ball sampling, where the purpose of the study was explained in advance, and consent and cooperation for participation in this study were obtained from the nine facilities, located in Hokkaido, Japan”.
Was it the result of a random sampling or simply included subjects were the ones meeting inclusion criteria and consenting to participate?
→Our study subjects were the ones meeting inclusion criteria and consenting to participate. Non-random sampling was a limitation of our study, so we added the following text to the limitations section.
Line 265: The number of cases of pneumonia onset was small, and a sampling bias among the facilities was found due to non-randomize sampling.
- What about selecting only 'elderly individuals with dementia'? Cognitive impairment has been robustly associated with bad oral health and this may represent a possible confounder in the analysis (doi: 10.3390/ijerph18136823)
→As you point out, cognitive impairment is strongly associated with poor oral health.
In order to minimize the influence of cognitive function, we sampled subjects using the degree of life independence, which was index that evaluates the degree of daily living due to dementia.
I added the following sentence and reference in lines 102, 370 and 376.
Line102:In order to minimize the influence of cognitive function, we sampled subjects using the degree of life independence, which is index that evaluates the degree of daily living due to dementia [31, 32].
The subject selection criteria included elderly individuals with dementia. The degree of life independence was evaluated as ≥ grade III for those who had a diagnosis of demen-tia, according to the guidelines of the Ministry of Health, Labour, and Welfare [33].
Line387:31. Maki Shirobe, Rena Nakayama, Hirohiko Hirano, Yuki Ohara, Keiko Endo, Yutaka Watanabe, Chiyoko Hakuta. Oral function and nutritional status among the elderly with facial and oral tactile hypersensitivity who are under long-term care. [Article in Japanese] 2017 Nihon Koshu Eisei Zasshi. doi: 10.11236/jph.64.7_351.
Line390:32. Mizue Suzuki, Hideyuki Hattori, Kunihiko Abe, Yuko Nakamura, Takayuki Saruhara. An analysis of the reliability and validity of the Life-trouble Scale for elderly patients living in geriatric facilities. [Article in Japanese]: 2018, Nihon Ronen Igakkai Zasshi. doi: 10.3143
- Line 120-122, please clarify this sentence.
I changed this sentence to the following sentence
Line108:“Exclusion criteria included individuals who had undergone dental therapeutic intervention during the year after the baseline survey in order to exclude the improvement of OAG score through dental treatment.”
- Since many readers are not familiar to OAG, please clarify which are the variables included in this score in materials and methods.
I added the following sentence in lines 80.
The OAG consists of eight assessment categories: voice, swallowing, lips, tongue, saliva, mucous membrane, gingiva, and teeth/dentures. Each category is expressed in three stages. The OAG score is the sum of the scores for each category; each category is scored from 1 to 3. The best possible score of 8 points is indicative of good oral health; the worst possible score of 24 points is indicative of poor oral health.
- Line 145-146, I don't understand whether OAG values were assessed using intraoral photographs. Please, clarify this point.
In order to avoid variations in OAG evaluation results, evaluation using OAG was performed in advance using existing intraoral photographs.
Inter-examiner ratings were adjusted based on discussion and training until a kappa coefficient of at least 0.9 was achieved.
- Please, report also how descriptive statistics were carried out.
We believe that the statistical processing performed is sufficiently described.
I would appreciate it if you could tell me about the items that should be added regarding statistical processing.
- Was a sample size calculation performed prior to subjects enrolment? I think that this point has a major impact on the validity and robustness of the conclusions.
We are planning a study of matched sets of subjects developing pneumonia or not. We calculated Sample size using Power and Sample size calculation (Version3.1.2). We could not be obtained the condition values for the calculation of the sample size from previous studies, so we assumed them and substituted into the calculation formula. a developing pneumonia among controls is 0.1 and the correlation coefficient for exposure between matched experimental and control subjects is 0.4. If the true odds ratio for failure in experimental subjects relative to control subjects is 2, we will need to study 266 experimental subjects with 1 matched control(s) per experimental subject to be able to reject the null hypothesis that this odds ratio equals 1 with probability (power) 0.9. The Type I error probability associated with this test of this null hypothesis is 0.05.
Results
- What reasons the drop-out were imputable? How did the authors manage a 25% drop-out rate in the analysis?
The causes of dropout are discharge from the facility, relocation, death, and refusal on the day of the survey. We suggested differences in dropout rates between facilities.
Dropout could be a selection bias because we were able not to control for dropout in our study.
I added the following text to the limitation section
“Our study was able not to control for dropout, so dropout could cause a selection bias”
- Please, provide descriptive statistics on the distribution of OAG scores among included subjects.
Thank you for your advice. We added to the distribution of OAG scores among included subjects in Table1 and Table 2.
- In general, the authors should make a greater effort to present results and tables in a friendlier format. This is too messy. In addition, Table 3 and 4 are more suitable for supplementary material than the main text.
Thank you for your advice. We revised the contents of the table and changed Tables 3 and 4 to supplemental data.
Round 2
Reviewer 2 Report
The authors have addressed all major changes and now the manuscript is suitable for final editing.
Author Response
(Referee 2): The authors have addressed all major changes and now the manuscript is suitable for final editing.
We thank you for your thoughtful suggestions.
We had the English proofread again by Language Editing Services (https://www.mdpi.com/authors/english).
I look forward to working with you to move this manuscript closer to publication in the IJERPH.
